# Towards equitable scheduling of global health teleconferences: a spatial exploration of the world's population and health by time zone

John A Crump ,[1] Tilman M Davies[2]

[1]Centre for International Health, University of Otago, Dunedin, Otago, New Zealand
[2]Department of Mathematics and Statistics, University of Otago, Dunedin, Otago, New Zealand

**Correspondence to**
Dr John A Crump;
john.crump@otago.ac.nz

## ABSTRACT

**Objectives** To examine the world's population, development, and health by condensed Coordinated Universal Time (UTC) offset to support a normative position on how to most defensibly schedule global health teleconferences with the primary goal of advancing equitable regard for participants by health condition.

**Design** Spatial exploration examined through the lens of equitable regard for participants.

**Setting** The Earth.

**Participants** The global population.

**Main outcome measures** Global population, countries, Human Development Index (HDI), and health measured in total and disease-specific disability-adjusted life years (DALYs) per 100 000 per year by UTC offset. Strengths and weaknesses of scheduling teleconferences accordingly to alternative approaches.

**Results** The UTC offset with the largest population of approximately 1 724 million persons occurred in UTC+5, largest country count of approximately 40 countries in UTC+1, and the lowest median HDI of 0.527 occurred in UTC0. The highest median total DALYs per 100 000 per year of 41 873 occurred in UTC+11, highest median HIV/AIDS DALYs per 100 000 per year of 941 in UTC0, and highest median typhoid and paratyphoid fevers DALYs per 100 000 per year of 279 occurred in UTC+5. Hypothetical teleconference scheduling scenarios generated temporal distances of up to 11 hours to UTC offsets representing the most countries and greatest number of DALYs per 100 000 per year. Teleconference scheduling based on egoism was considered to be problematic, and contractualism as unrealistically demanding. Utilitarianism resulted in the risk of systematically disadvantaging small, temporally remote groups. Egalitarianism offered equality, but prioritarianism offered the possibility of addressing health inequities.

**Conclusions** Global health teleconferences may generate great temporal distances between participants compromising representative participation, as well as the well-being of attendees. Our spatial exploration of the world's population and health by UTC offset may be used to support a normative position on scheduling global health teleconferences that enhance progress towards health equity.

## STRENGTHS AND LIMITATIONS OF THIS STUDY

⇒ We displayed the world's population and health by time zone to inform a normative position on scheduling global health teleconferences that span many or most global time zones.

⇒ Inequities in teleconference scheduled were illustrated with hypothetical scenarios through calculating temporal distance between two participants in relation to largest population and country counts, the lowest median Human Development Index, and highest median total and disease-specific disability-adjusted life year rate Coordinated Universal Time offsets.

⇒ An ethics assessment of the strengths and weaknesses of alternative strategies for scheduling international teleconferences in global health was made in the context of ethics principles of distribution, examining strategies based on egoism, contractualism, utilitarianism, egalitarianism, and prioritarianism.

## INTRODUCTION

Addressing global health problems through collaboration depends in part on international synchronous communication and coordination.[1] In modern times, this is typically achieved by teleconference, a collective term for communication using a range of platforms including voice only conference calls, videoconferences, and web-based meetings. Addressing the COVID-19 pandemic has at once demanded increased levels of global health communication and coordination, and driven restrictions on face-to-face meetings,[2] resulting also in a marked increase in demand for international teleconferences for communication on a wide range of health conditions. While teleconferences have been considered inferior to face-to-face meetings for complex tasks with high ambiguity and where new ties need to be built,[3] the aspiration to pursue more judicious use of cleaner international air travel following the

COVID-19 pandemic[4] point to a future with a sustained high level of international teleconference use.

Since the Earth approximates a sphere that orbits the sun and spins, daylight is experienced in sequence by longitude on a 24-hour cycle. To manage this, the world is presently organised into 37 longitudinal time zones adapted to follow national and subnational boundaries, and observed as Coordinated Universal Time (UTC) offsets from the prime meridian in Greenwich, England, the international standard for 0° longitude,[5] and home to UTC offset zero or 'zulu time.' The value of being home to the prime meridian was reflected in the 'scramble' among predominantly western nations to become the host, a matter that was resolved at a conference in Washington, DC, in the USA in 1884.[5] Among the many consequences of this decision experienced today, participants joining teleconferences at locations remote to zulu time may be more likely to be asked to participate between the hours of 23:00 through 5:00 local time, a time window classified as unfavourable by some online teleconference scheduling tools, and a manifestation of a form of discrimination termed antimeridianism.[6]

Temporal distance is a measure, distinct from geographic distance, defined as a directional measurement of the temporal displacement experienced by two collaborators who want to interact with each other.[7] Because humans are a diurnal species with a circadian clock entrained closely to light, traditionally from the sun,[8] the temporal distance inherent in synchronous multiple time zone teleconferencing poses a challenge to representative participation[9] and to supporting the well-being of attendees.[10] Methods in common use for scheduling international conference calls include online tools facilitating identification of times convenient to the greatest number of potential participants, a form of utilitarianism. However, scheduling may be influenced by other drivers including egoism, where the preferences of the organiser are prioritised. In teleconference spanning many or most global time zones, some participants will inevitably be asked to join between the hours of 23:00 through 5:00 local time. But who should be asked to join at these times and why? While global health claims to seek health equity,[1] it is recognised that aspects of its history in colonial medicine, tropical medicine, and international health may perpetuate inequities.[11] The decolonising global health movement 'fights against ingrained systems of dominance and power in the work to improve the health of populations, whether this occurs between countries, including between previously colonising and plundered nations, and within countries.'[12] To our knowledge, there has hitherto been little specific attention given to supporting equity in the organisation and scheduling of international teleconferences in global health.

In order to inform equitable approaches to scheduling multiple time zone teleconferences in global health, we sought to describe empirical data on the world's population and its health by time zone. The primary objectives of this analysis were to display human population characteristics and health by time zone and to consider the implications of these findings in the context of the ethics of distribution, an area of ethics that addresses the distribution of benefits and burdens across separate persons or groups. We aspired to identify a normative position to enhance equitable organisation and participation, and ultimately outcomes, for teleconferences in global health.

## METHODS

We explored the world's population and its health by time zone. We illustrated the implications of the findings with worked examples, and assessed alternative strategies for international multiple time zone teleconference scheduling in light of the ethics of distribution, with the primary goal of advancing equitable regard for participants by health condition.

### Time zones

We constructed a fine grid of longitude coordinates ranging from −180° to 180° and of latitude coordinates ranging from −90° to 90° and fed the grid into the lutz ('look up time zones') software[13] in the R programming environment.[14] Each coordinate was linked to a country and region identifier, and to a non-daylight savings time UTC offset value. For simplicity, fractional UTC offsets were 'shrunk' towards zero. For example, UTC+8.5 translated to UTC+8 and UTC−3.5 to UTC−3. For countries with multiple time zones, their contributing UTC values were first averaged, and then the shrinking translation was performed. Multiple time zone country's data were assigned to that single shrunk time zone. A small number of countries with UTC offsets greater than +12 were shrunk to +12. Following the averaging operation and shrinking translation, each country was allocated one of the resulting 25 'simplified' UTC zones ranging from UTC−12 through zulu time to UTC+12.

### World population, countries, and development metrics

We used the United Nations World Population Prospects adjusted version of the 2020 population count data.[15] Each of the simplified global UTC offsets was then converted to a two-dimensional irregular polygon describing the shape of the area of the UTC offset, and the approximate population count in each zone was obtained by cross-referencing the global population count data within these polygons. We summed together the contributing areas of the global population count map within each simplified UTC offset polygon. We approximated country counts using the centroids of each country's geographic region cross-referenced with the simplified UTC polygons.[16 17]

The Human Development Index (HDI) is a national composite index using the geometric mean of scores for health assessed by life expectancy at birth, education measured by mean of years of schooling for adults aged ≥25 years and expected years of schooling for children of school entering age, and standard of living measured by

gross national income per capita. We obtained national HDI scores for the year 2019 from the United Nations Development Programme.[18] HDI scores were categorised as >0.800: very high; 0.700–0.799: high; 0.550–0.699: medium; and <0.550: low. All country-specific values such as HDI were cross-referenced with our country-to-simplified-UTC offset dataset.

## Health metrics

Disability-adjusted life years (DALYs) are a societal measure of the burden of disease experienced by a population. DALYs are calculated as the sum of years of life lost from premature death and years lost due to disability. Years lost due to disability are calculated by multiplying disease incidence by the disability weight, a measure of severity for the condition, by the duration of illness. DALYs per 100 000 for the year 2019, and for the illustrative diseases HIV/AIDS and typhoid and paratyphoid fevers, were accessed through the Institute for Health Metrics and Evaluation Global Health Data Exchange.[19] DALY data were cross-referenced with our country-to-simplified-UTC offset dataset.

## Data visualisation

A world map was created in geographic coordinates using the World Geodetic System 1984 datum with colour-coded UTC offsets. UTC offset colour-coded bar plot panels were generated for each population and health metric corresponding to the map. In addition, an interactive three-dimensional UTC split-plot was created.

## Temporal distance worked examples

To illustrate the impact of teleconference scheduling on temporal distance for two participants, we calculated temporal distance from a hypothetical call scheduled at noon in London, UK (UTC0); noon in Washington, DC, USA (UTC−5); and noon in New Delhi, India (UTC+5), to the largest population and country counts, the lowest median HDI, and the highest median DALY rate UTC offsets identified through our spatial exploration.

## Ethics framework

To support a normative position, we considered the strengths and weaknesses of alternative strategies for scheduling international global health teleconferences in the context of ethics principles of distribution, an area of ethics that addresses how the benefits and burdens are distributed across separate persons or groups. These principles included egoism, which prioritises self-interest;[20] contractualism that holds an act to be wrong if its performance under the circumstances would be disallowed by any set of principles for the general regulation of behaviour that no one could reasonably reject as a basis for informed, unforced, general agreement;[21] utilitarianism the view that the morally right action is the action that produces the most good;[22] egalitarianism, a school of thought that favours equality of some sort;[23] and prioritarianism, a view that the goodness of an outcome is a function not only of the overall well-being across all individuals, but with extra weight given to worse-off individuals.[24]

## Patient and public involvement

Our research was informed by two decades of global health teleconference participation, including from a low-income and middle-income country perspective in Tanzania, and from the perspectives of participation from New Zealand in UTC+12. Interest in the topic was intensified during COVID-19 pandemic travel restrictions.

# RESULTS

## World population, countries, and development by time zone

We demonstrated that the simplified UTC offset with the largest population of approximately 1724 million persons occurred in UTC+5, followed by UTC+8 at 1630 million persons, and UTC+1 at 852 million persons. The largest country count of approximately 40 countries occurred in UTC+1, followed by UTC+2 at 26, and UTC+3 at 22. Of 203 countries classified, 20 (9.9%) were in time zones of Oceania, ranging from UTC+10 to UTC−7. The lowest median HDI of 0.527 occurred at UTC0, followed by UTC+11 at 0.588, and UTC+10 at 0.620 (figure 1 and table 1).

## World health by time zone

The UTC offset with the highest median total DALYs per 100 000 per year of 41 872 occurred in UTC+11, followed by UTC0 at 37 563, and UTC+10 at 37 489. The UTC offset with the highest median HIV/AIDS DALYs per 100 000 per year of 941 occurred in UTC0, followed by UTC−2 at 898, and UTC−5 at 696. By contrast, the UTC offset with the highest median typhoid and paratyphoid fevers DALYs per 100 000 per year of 279 occurred in UTC+5, followed by UTC0 at 133, and UTC+7 at 119 (figure 1 and table 1). An interactive three-dimensional plot of world population and health data by UTC offset is available at: https://www.stats.otago.ac.nz/~tdavies/utcsplit_health.html (allow up to 30 s to load in browser) and a static image is available as an online supplemental figure 1.

## Temporal distance worked examples

Table 2 shows temporal distances generated between two participants by hypothetical teleconferences scheduled for noon local time in London, UK; Washington, DC, USA; and New Delhi, India, to the identified UTC offset with the largest total population and largest total country count, the lowest median HDI, and the highest total and disease-specific DALY rates. These simple, two-participant scenarios generated temporal distances of up to 11 hours to UTC offsets representing the largest country count and highest median DALYs per 100 000 per year.

## Ethics assessment

Table 3 summarises the strengths and weaknesses of alternative strategies for scheduling international

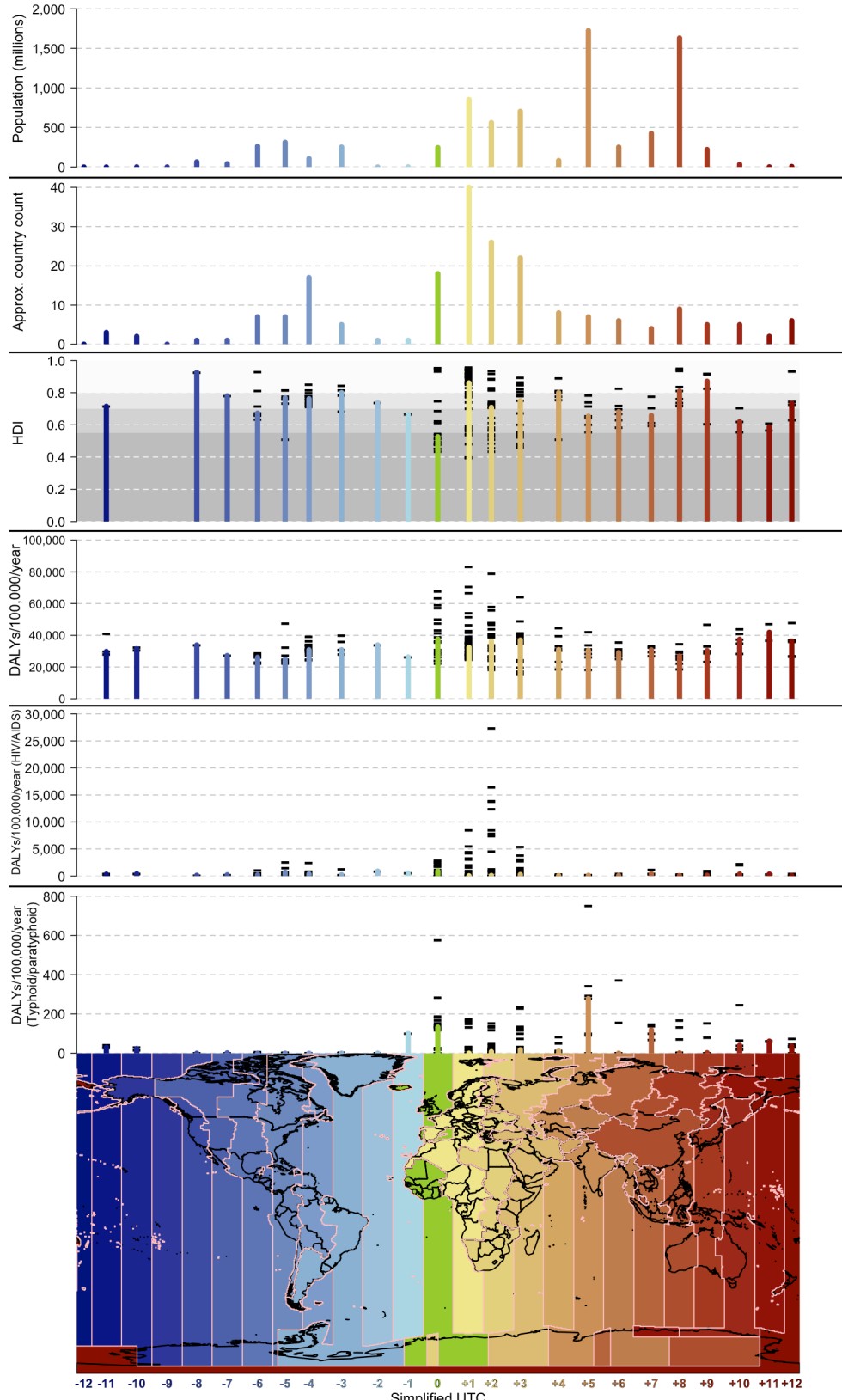

**Figure 1** Global population, countries, HDI, total and disease-specific disability-adjusted life years per 100 000 by simplified UTC offset, 2019–2020. Simplified UTC offsets are colour coded with pink lines at boundaries. National borders are represented with black lines. Plot bars show totals for population and country count. Plot bars show median values for HDI and DALYs; black horizontal lines represent individual scores of each contributing country. Greyscale shading on the HDI plot represents categories of human development used by the United Nations Development Programme: >0.800: very high; 0.700–0.799: high; 0.550–0.699: medium; and <0.550: low. DALYs, disability-adjusted life years; HDI, Human Development Index; UTC, Coordinated Universal Time.

**Table 1** Data summary of global population, countries, Human Development Index, total, and disease-specific disability-adjusted life years per 100 000 per year by simplified Coordinated Universal Time offset, 2019–2020

| Characteristics | Coordinated universal time offset | | | | | | | | | | | |
| --- | --- | --- | --- | --- | --- | --- | --- | --- | --- | --- | --- | --- |
| | −12 | −11 | −10 | −9 | −8 | −7 | −6 | −5 | −4 | −3 | −2 | −1 |
| Population | | | | | | | | | | | | |
| Population, millions | 0.00 | 0.08 | 1.80 | 0.63 | 64.35 | 40.43 | 261.90 | 311.67 | 106.33 | 252.00 | 0.14 | 1.60 |
| Country count | 0 | 3 | 2 | 0 | 1 | 1 | 7 | 7 | 17 | 5 | 1 | 1 |
| HDI, median | – | 0.715 | – | – | 0.926 | 0.779 | 0.673 | 0.767 | 0.762 | 0.800 | 0.738 | 0.665 |
| Health | | | | | | | | | | | | |
| DALY rate, total | – | 29768 | 31485 | – | 33866 | 27197 | 26108 | 24875 | 30834 | 30884 | 33899 | 26307 |
| DALY rate, HIV/AIDS | – | 333.37 | 429.95 | – | 126.63 | 211.86 | 216.98 | 696.45 | 379.01 | 227.17 | 898.37 | 522.57 |
| DALY rate, typhoid and paratyphoid | – | 30.79 | 24.11 | – | 0.02 | 1.94 | 0.29 | 0.35 | 0.12 | 0.08 | 0.70 | 100.00 |

| Characteristics | Coordinated universal time offset | | | | | | | | | | | |
| --- | --- | --- | --- | --- | --- | --- | --- | --- | --- | --- | --- | --- |
| | 0 | 1 | 2 | 3 | 4 | 5 | 6 | 7 | 8 | 9 | 10 | 11 | 12 |
| Population | | | | | | | | | | | | | |
| Population, millions | 244.65 | 851.90 | 545.55 | 716.67 | 81.13 | 1723.74 | 251.18 | 424.72 | 1630.13 | 221.12 | 32.81 | 2.14 | 6.49 |
| Country count | 18 | 40 | 26 | 22 | 8 | 7 | 6 | 4 | 9 | 5 | 5 | 2 | 6 |
| HDI, median | 0.527 | 0.860 | 0.709 | 0.750 | 0.800 | 0.654 | 0.683 | 0.659 | 0.810 | 0.871 | 0.620 | 0.588 | 0.734 |
| Health | | | | | | | | | | | | | |
| DALY rate, total | 37563 | 32428 | 36295 | 36965 | 30970 | 30538 | 29078 | 30841 | 26871 | 30353 | 37489 | 41872 | 36348 |
| DALY rate, HIV/AIDS | 940.60 | 36.38 | 85.63 | 272.46 | 66.42 | 117.70 | 87.94 | 416.09 | 69.23 | 143.40 | 369.73 | 350.74 | 93.10 |
| DALY rate, typhoid and paratyphoid | 132.98 | 0.06 | 8.05 | 17.34 | 10.89 | 278.90 | 0.39 | 118.96 | 1.36 | 2.85 | 40.72 | 61.45 | 36.17 |

DALY rate, disability-adjusted life years per 100 000 per year; HDI, Human Development Index; UTC, Coordinated Universal Time.

**Table 2** Temporal distance generated in hypothetical teleconferences to key population and health Universal Coordinated Time offsets

| Host city | Local time | UTC offset | Temporal distance, hours* | | | | | | |
|---|---|---|---|---|---|---|---|---|---|
| | | | Largest total population | Largest country count | Lowest median HDI | Highest median total DALY rate | Highest median HIV/AIDS DALY rate | Highest median typhoid and paratyphoid DALY rate |
| Washington, DC, USA | Noon | −5 | 10 | 6 | 5 | 9 | 5 | 10 |
| London, UK | Noon | 0 | 5 | 1 | 0 | 11 | 0 | 5 |
| New Delhi, India | Noon | +5 | 0 | 4 | 5 | 6 | 5 | 0 |

*Shortest temporal distance in either easterly or westerly direction.
DALY rate, disability-adjusted life years per 100 000 per year; HDI, Human Development Index; UTC, Coordinated Universal Time.

teleconferences in global health in the context of ethics principles of distribution. Egoism was identified as problematic and contractualism as unrealistically demanding for large teleconferences. Utilitarianism resulted in the potential for systematic disadvantage to small, temporally remote groups. Egalitarianism offered equality, but prioritarianism with its emphasis on offering explicit priority or precedence to disadvantaged or affected groups offered the possibility of addressing inequities.

## DISCUSSION
### Statement of principal findings
We demonstrate that the largest global population, largest country count, lowest median HDI, and the highest median total DALYs per 100 000 per year, and highest median disease-specific DALYs per 100 000 per year span a wide range of UTC offsets that can yield large temporal distances between international teleconference participants. In international teleconferences spanning many or most of the world's time zones, it is inevitable that some participants will be asked to join during the unfavourable window 23:00 through 5:00 local time. But who should be asked to join during those hours and on what basis? In order to optimise representative participation and to support the well-being of attendees, we identify that teleconference scheduling based on egoism, and even utilitarian strategies that consistently favour the majority, is problematic. We suggest that at minimum egalitarian, and ideally prioritarian, approaches offering explicit priority or precedence to disadvantaged or affected groups be adopted in most circumstances.

### Strengths and weaknesses of the study
To our knowledge, ours is the first attempt to display the world's population and health by time zone to inform teleconference scheduling in global health. While unique, our study had a number of limitations. We were unable to identify robust empiric data on the actual scheduling of global health conference calls and the impacts on participation and participants, so the premise for biases

**Table 3** Ethics of distribution considerations in global health teleconference scheduling

| Approach, reference | Summary | Scheduling priority | Strengths | Weaknesses |
|---|---|---|---|---|
| Egoism[20] | Maximise self-interest | Self-convenience | Limits disruption to the organiser | Compromises participation and well-being of invitees |
| Contractualism[21] | Wrong if disallowed by any set of principles for general regulation of behaviour | Self-regard and respect for others | Allows a variety of individual reasons to be considered | Highly demanding to implement, especially in large groups with weak ties |
| Utilitarianism[22] | Right action produces the most good | Convenient to the majority | Optimises the number of participants | Systematically disadvantages minority populations |
| Egalitarianism[23] | Favours equality of some sort | Inconvenience shared | Provides equal convenience and opportunity to participate | Insufficient to provide equity to disadvantaged groups |
| Prioritarianism[24] | Well-being across all individuals, with extra weight given to worse-off individuals | Priority to most affected, most hard-pressed, or least represented | Offers equity for disadvantaged groups | May yield fewer participants overall |

in teleconference scheduling was based on the qualitative experience of the study team and public involvement garnered through fieldwork in a low-income and middle-income country and in the UTC+12 offset. We acknowledge that not all humans prefer to work during daylight hours, or to recreate. Because some time zones are offset from UTC by fractions of hours, at the time of writing there were 37 local times in use worldwide. For ease of presentation and interpretation, we chose to 'shrink' time zones representing fractions of hours towards zero resulting in conflation of the 37 time zones to UTC and its 24 offsets. While this resulted in loss of 13 fractional time zones, we think that our approach reasonably approximated local apparent solar time, and thus corresponds well with the diurnal lifestyle of humans. Similarly, since countries with multiple time zones had contributing UTC values for population health metrics first averaged and then 'shrunk,' multiple time zone countries are represented in only single time zones. We considered that the resulting 'bunching' of metrics into single time zones was preferable to counting individual countries more than once. We recognise that UTC offset is an imperfect surrogate for local apparent solar time, a superior but less practical benchmark for measuring inconvenience in relation to the human circadian cycle. Further, population and health metrics by UTC offset are approximations based on limitations of source data and subsequent translations. Care should be taken when interpreting median values from UTC offsets that represent few countries, such as UTC+11 dominated by the Solomon Islands and Vanuatu. However, we do not anticipate that this had major impacts on rankings by UTC offset. We also acknowledge that factors beyond those contributing to equity of participation may be relevant in teleconference scheduling, but explicit consideration of these was beyond the scope of our analysis.

## Comparison with other studies

We are not aware of similar research examining the world's population, countries, development, and health by UTC offset, nor of work attempting to support a normative position on how to most defensibly schedule global health teleconferences with the primary goal of advancing equitable regard for participants by health condition. Nonetheless, our analysis contributes to broader discourse on the decolonisation of global health.[11 12] Coloniality in global health teleconferencing may be evident when calls on health conditions that most affect 'previously plundered nations' are led by, or scheduled to the preferences of, those in 'previously colonising nations' of the 'Global North.'[12 25] In the hypothetical worked examples, a prioritarian approach to scheduling a teleconference focused on HIV/AIDS would favour the highest global HIV/AIDS DALY rate offset at UTC0. However, scheduling a teleconference on typhoid fever and paratyphoid fever based on egoism favouring the self-convenience of participants in Washington, DC, USA, would create a large temporal distance of 10 hours to the highest global typhoid and paratyphoid fever DALY rate offset at UTC+5. By contrast, temporal distance to those most affected by typhoid and paratyphoid fevers would be dramatically reduced by a prioritarian approach favouring daylight hours in New Delhi, India. Our spatial exploration by UTC offset could be extended to other diseases using publicly available datasets.

## Implications for clinicians and policy-makers

Persons asked to join international teleconference between 23:00 and 5:00 may be less able or inclined to participate and thus be tacitly systematically excluded from global health discourse, a manifestation of a form of discrimination termed antimeridianism. Such participants may experience circadian disruption known to have unfavourable impacts on well-being.[10] Decision-making on who should be inconvenienced may be considered an application of the ethics of distribution. It would seem that there are few situations when egoism, when expressed as self-interest based on self-convenience, should be the primary driver of global health teleconference scheduling, as representative participation and global health goals are compromised. Contractualism, while attractive as a means to incorporate a range of personal preferences, is highly demanding. For example, both self-regard and respect for others are required, and these may be in doubt when social ties are weak in large global health teleconferences that are not founded in established relationships.[3] Used without nuance, utilitarian methods using online scheduling tools designed to identify times convenient to the greatest number of potential participants may systematically disadvantage minority groups representing less populous UTC offsets, such as those of Oceania. This may be of particular concern if critical participants reside in these offsets, including those highly affected by the teleconference health condition of focus.

Egalitarian approaches to teleconference scheduling include systematically rotating UTC offsets to ensure that the burden of inconvenient times, or the ability to reasonably participate, is shared equally among attendees. An alternative egalitarian strategy is to adopt partial asynchrony, holding the same teleconference more than once at different UTC offsets. However, partial asynchrony compromises efficiency and precludes simultaneous, whole-group interaction. Whereas equality provides each individual or group of people the same resources and opportunities, equity recognises that each person has different circumstances, and allocates the resources and opportunities needed to reach an equal outcome.[26] Since global health strives for health equity,[1] we suggest that prioritarian approaches offering explicit priority or precedence to disadvantaged or affected groups should be strongly considered in international teleconference scheduling. We acknowledge that this approach, favouring one group over others, may

appear counterintuitive ethically. However, in this instance, it seeks to benefit those who are central to the health condition under investigation, and hence support their interests.

Finally, there are many domains of global health in which moral rhetoric is not matched by practice.[27] We hope that our pragmatic analysis and normative position might be an exception and that our findings can be translated in better practice in teleconference scheduling by clinicians and policy-makers in the global health field.

## CONCLUSION

In conclusion, while just one of a range of fundamental moral and ethical inconsistencies in global health, teleconferences may generate great temporal distances between participants that can compromise representative participation, as well as the well-being of attendees. In international teleconferencing spanning many or most of the world's time zones, some participants will inevitably be asked to participate between 23:00 and 5:00 local time. While egalitarianism should be considered a minimum standard, our spatial exploration of the world's population and its health by UTC offset can be used to inform prioritarian approaches to scheduling international teleconferences, offering explicit priority or precedence to disadvantaged or affected groups, in turn enhancing the global health aspiration to strive for health equity. We are developing practical tools to display global population, countries, Human Development Index, and health metrics by condition by UTC offset to inform global health teleconference scheduling.

**Acknowledgements** We thank Emeritus Professor D. Gareth Jones, MB BS, DSc, MD, Bioethics Centre, University of Otago, Dunedin, New Zealand, for critical review of the manuscript, focussing on the ethics framework and assessment. We acknowledge the use of several public data sources for this project including the NASA Socioeconomic Data and Applications Center gridded population of the world version 4 for population density through Creative Commons Attribution international license 4.0 (https://creativecommons.org/licenses/by/4.0/); the United Nations Population Division for population and country data, and the United Nations Development Programme human development data center for human development index data, both through Creative Commons Attribution intergovernmental licence 3.0 (https://creativecommons.org/licenses/by/3.0/igo/legalcode); and the Institute for Health Metrics and Evaluation for health data through Creative Commons 4.0 attribution-non-commercial international licence 4.0 (https://creativecommons.org/licenses/by-nc/4.0/).

**Contributors** JAC conceived the work. JAC and TMD designed the analysis. TMD undertook spatial exploration and created the figures. JAC did the temporal distance calculations and made the ethics assessment. JAC wrote the first draft of the manuscript, and TMD critically revised the manuscript for important intellectual content and gave final approval of the version to be published. JAC and TMD are the guarantors. The corresponding author attests that both listed authors meet authorship criteria and that no others meeting the criteria have been omitted.

**Funding** JAC received support from Bill & Melinda Gates Foundation grant (OPP1151153).

**Competing interests** None declared.

**Patient and public involvement** Patients and/or the public were involved in the design, or conduct, or reporting, or dissemination plans of this research. Refer to the Methods section for further details.

**Patient consent for publication** Not applicable.

**Ethics approval** Not applicable.

**Provenance and peer review** Not commissioned; externally peer reviewed.

**Data availability statement** Data are available upon reasonable request. Dataset, statistical code, and analytical methods are available via request to the corresponding author to other researchers for purposes of reproducing the results, replicating, or extending the analysis. The lead author JAC affirms that the manuscript is an honest, accurate, and transparent account of the study being reported; that no important aspects of the study have been omitted; and that any discrepancies from the study as planned have been explained.

**ORCID iD**
John A Crump http://orcid.org/0000-0002-4529-102X

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
