## [Reviewer comments · BMJ Open]

ARTICLE DETAILS

TITLE (PROVISIONAL)	Towards equitable scheduling of global health teleconferences: a spatial exploration of the world's population and health by time zone
AUTHORS	Crump, John; Davies, Tilman

VERSION 1 – REVIEW

REVIEWER	Sayeed, Sadath Harvard University
REVIEW RETURNED	04-Oct-2021

GENERAL COMMENTS	This is an original investigation that creatively utilizes objective data about the world's population, burden of disease and time zones to make a normative claim about the fairness of teleconferencing scheduling when the primary premise of such conferencing is to advance health equity. The primary strength of the manuscript is its unique data transposition of key population-level health outcomes to relative time zones. The results of this "spatial exploration" (authors' terms) are not surprising given that it tracks with what is well known about burden of disease. The primary weakness of the manuscript is in its development of a complete normative argument. It is commonly understood/assumed that those in positions of relative power (i.e. access to relevant resources) control and dictate the terms of engagement in global health. Indeed, the concept of "global health" in a basic operational sense depends on resource rich individuals, institutions, and governments caring enough in the first place about the less fortunate. The authors are entirely correct to call out those who control and dictate what global health is, what problems should be addressed, etc. that equity is not at all driving how agendas are being set or how problems are being solved - regardless of the pleasing rhetoric we repeatedly are exposed to by the same. There is a basic moral/ethical inconsistency (some would say hypocrisy) in how global health is conducting itself to the extent that its leaders/proponents insist that health outcome equity is the primary goal/aim, and this continues to be downplayed by those in power for problematic reasons. The authors' well-grounded (if anecdotal) intuition that global health teleconferences (and live conferences) by and large cater to the well-being of the already well-positioned and privileged actors is not only obvious, it is symptomatic of a much deeper set of uncomfortable truths - global health for the most part remains a charitable hobby of the rich and powerful who spend more time and resources on serving their interests than delivering sustained health care justice and/or equity to people in desperate need. The fashionable multi-disciplinary domain of global health, and its many antecedents, has a long track record of discounting the voices of those it claims to most concerned about; this is too often what self-congratulatory, increasingly reified expertise perpetuates.
---

	The authors' treatment of the substantial normative questions/issues in this paper is simplistic and somewhat formulaic. This is entirely understandable in so far as neither (professionally) self-identifies as an expert in moral or political philosophy or ethics. Nevertheless, to the extent the manuscript insists on making a demanding normative conclusion/claim, much more work could be done here to fully substantiate why a prioritarian model merits consideration over other ethical constructs. Consider a simple example in which by having the teleconference follow a self-serving/convenience model that caters to those with the most resources in turn generates substantially more resources to utilize in global health that then go on to actually deliver tangible health benefits to the populations in need - is health equity in the more important sense advanced by adopting this scheme? Put differently, even if the participants/population of interest are inconvenienced and somewhat excluded from the meeting because of scheduling, but later are massively rewarded with resources they otherwise would not have obtained because the relevant stakeholders would not have attended a meeting that inconvenienced them, is this an ethically inferior outcome? In reality, this is a complicated ethical calculus, and more degrees of granularity would significantly improve the initial commendable effort made by the authors. These problems in normative analysis, however, are not fatal to the manuscript in this reviewer's opinion. Ideally, a manuscript such as this further opens the minds of many privileged actors in the global health space to recognizing just how saturated through our work is with increasingly institutionalized inequity, and that radical reformation is called for if we want to avoid growing, legitimate indictments of moral hypocrisy. Specific suggestions for revision: This reviewer found the abstract confusing to understand. The term "ethics of distribution" is vague and distinctly unhelpful as a simplified analytical construct to engage the reader; it needs elaboration and/or more precision in terms of what the authors' are trying to convey as a framework to guide their normative analysis. The abstract could simply state that the paper uses a specific, novel methodological strategy to map condensed time zones onto population health metrics to support a normative claim about how we might "most defensibly" schedule global health conferences where advancing equitable regard/treatment of would-be condition, relevant participants is the primary aim/goal. If the authors are interested in developing their normative analysis, and/or calling out a rich tradition of moral inconsistency in terms of the lofty moral rhetoric rarely matching the actual work within global health, that would also be most welcome.
--	--

REVIEWER	Tariku, Binyam College of Health Science Dilla University, Department of Health Informatics
REVIEW RETURNED	29-Dec-2021

GENERAL COMMENTS	This paper describes approaches of scheduling teleconferences for multiple time zones, which is an important notion to address global health issue. I believe study findings will provides an important stepping stone for alleviating the accessibility and equitability global telehealth conferences. However, it will be great if the authors includes implication and recommendation for future studies.
--

REVIEWER	Lipton, Robert Prevention Research Center
REVIEW RETURNED	18-Jan-2022

GENERAL COMMENTS	I really like this paper as it does something different in exploring disparities/inequities, globally. I particular like the inclusion of ethical considerations as well as some very interesting/novel graphics, particularly the globe! The DALY approach is also important. My sense, as well, is this is a bit of a trojan horse for larger, important disparities work and begs the question about focusing on specific cases, like comparing India to the UK or the US. And what do we do about Russia - Twelve time zones averaged into one!?!? I think this is a good first pass at dealing with such issues using this spatial approach but as it stands, It's a bit vulnerable to over averaging – ecological fallacy issues, to some extent. Also, in terms of the ethics discussion, it either needs more or less discussion, as does the colonized versus colonizer framework. Is there a hierarchy of ethical frameworks? Or do we really need to just default to egalitarian approaches, notwithstanding it may be “Insufficient to provide equity to disadvantaged groups” all things being equal. (btw, isn't this almost always going to be the case in this very imperfect world?). Also, what if the “center” (center versus periphery) contribution is absolutely essential to helping mitigate health disparities and thus should be accorded some importance (again, this is both in the realm of philosophical discussion and the reality of doing international health). Another issue, which is beyond the scope of this work has to do with the possibility that even with the temporal inequity, such teleconferences might be better than the alternatives. Some discussion of this may be appropriate. Also, clarification between conferences and meetings should be made and a brief discussion of costs and benefits. Another interesting issue is that the authors mention two decades of data collection which seems like something really important and demands a bit more explanation. We can imagine all kinds of really important information coming out of this project. Small things: the shift worker – carcinogen connection seems a bit overstated for this work. Not really needed to make the larger argument. I do think the first figure with the timeline and world atlas needs to be higher resolution and/or larger, as it stands, it's a bit unreadable but very interesting
--

VERSION 1 – AUTHOR RESPONSE

Reviewer: 1
Dr. Sadath Sayeed, Harvard University

1. Comment: This is an original investigation that creatively utilizes objective data about the world's population, burden of disease and time zones to make a normative claim about the fairness of teleconferencing scheduling when the primary premise of such conferencing is to advance health equity. The primary strength of the manuscript is its unique data transposition of key population-level health outcomes to relative time zones. The results of this "spatial exploration" (authors' terms) are not surprising given that it tracks with what is well known about burden of disease.

Response: The authors thank the review for the positive feedback. We chose the term 'spatial exploration' for our exploratory analysis of descriptive data since our use of spatial data does not meet the threshold for being considered a formal geospatial analysis.

2. Comment: The primary weakness of the manuscript is in its development of a complete normative argument. It is commonly understood/assumed that those in positions of relative power (i.e. access to relevant resources) control and dictate the terms of engagement in global health. Indeed, the concept of "global health" in a basic operational sense depends on resource rich individuals, institutions, and governments caring enough in the first place about the less fortunate. The authors are entirely correct to call out those who control and dictate what global health is, what problems should be addressed, etc. that equity is not at all driving how agendas are being set or how problems are being solved - regardless of the pleasing rhetoric we repeatedly are exposed to by the same. There is a basic moral/ethical inconsistency (some would say hypocrisy) in how global health is conducting itself to the extent that its leaders/proponents insist that health outcome equity is the primary goal/aim, and this continues to be downplayed by those in power for problematic reasons. The authors' well-grounded (if anecdotal) intuition that global health teleconferences (and live conferences) by and large cater to the well-being of the already well-positioned and privileged actors is not only obvious, it is symptomatic of a much deeper set of uncomfortable truths - global health for the most part remains a charitable hobby of the rich and powerful who spend more time and resources on serving their interests than delivering sustained health care justice and/or equity to people in desperate need. The fashionable multi-disciplinary domain of global health, and its many antecedents, has a long track record of discounting the voices of those it claims to most concerned about; this is too often what self-congratulatory, increasingly reified expertise perpetuates.

Response: The authors thank the reviewer for the carefully considered and highly insightful comments about wider structural concerns within the field of global health. We agree that global health is symptomatic of, and perpetuates, a range of fundamental moral and ethical inconsistencies. In this context our narrow focus on teleconference scheduling might seem trivial, if not just another symptom of much wider and pervasive uncomfortable truths that we attempted to touch on in the Introduction. That said, as residents of UTC+12 we consider that we are particularly well positioned to highlight inequities in teleconference scheduling. While not emphasized heavily in our paper, the 'scramble to host the prime meridian' finally resolved at the International Conference held in October 1884 at Washington was dominated by representatives from countries of Europe and the Americas. The countries most negatively impacted by the decision to place the prime meridian in Greenwich, England, had no voice at the meeting, yet the consequences of that decision underpin at least part of what we are trying to articulate in our analysis. Indeed, because time zones are arranged longitudinally, they cross the Brandt Line separating the so-called 'Global South' from the 'Global North.' As such, problems with teleconference scheduling are not simply those of rich individuals in wealthy nations controlling the poor. After all, a teleconference scheduled to the convenience of someone in Geneva, Switzerland, is likely also to also be convenient to residents of any number of African countries. The concern is that teleconferences routinely scheduled for the convenience of someone in Geneva, Switzerland will systematically inconvenience or, more concerning, exclude those resident in UTC offsets remote to Geneva, often without consideration for 'need.' It was our observation over many years that participants resident at UTC offsets remote to the prime meridian are routinely requested to join calls 11pm through 5am local time independent of poverty level or disease burden, a time window classified as unfavorable by some online teleconference scheduling tools. One consequence of this is that persons and populations remote to zulu time are systematically underrepresented in global health discourse, irrespective of wealth or health metrics. As we point out in our paper, UTC+11, while home to a relatively small populations of the Solomon Islands and Vanuatu, carries the highest per capita burden of premature illness and death measured by disability adjusted life years, and experiences the second lowest human development index score globally. All of this said, we have taken the reviewer's excellent points to heart and have bolstered content on the broader context of colonialism in global health in the Introduction. Further, we now return to the wider context in the Conclusion of the Discussion. We have also expanded our treatment of 'the scramble to host the prime meridian,' since this relevant history and its consequences may not be widely appreciated by those who are temporally proximal to zulu time.

2. Comment: The authors' treatment of the substantial normative questions/issues in this paper is simplistic and somewhat formulaic. This is entirely understandable in so far as neither (professionally) self-identifies as an expert in moral or political philosophy or ethics. Nevertheless, to

the extent the manuscript insists on making a demanding normative conclusion/claim, much more work could be done here to fully substantiate why a prioritarian model merits consideration over other ethical constructs. Consider a simple example in which by having the teleconference follow a self-serving/convenience model that caters to those with the most resources in turn generates substantially more resources to utilize in global health that then go on to actually deliver tangible health benefits to the populations in need - is health equity in the more important sense advanced by adopting this scheme? Put differently, even if the participants/population of interest are inconvenienced and somewhat excluded from the meeting because of scheduling, but later are massively rewarded with resources they otherwise would not have obtained because the relevant stakeholders would not have attended a meeting that inconvenienced them, is this an ethically inferior outcome? In reality, this is a complicated ethical calculus, and more degrees of granularity would significantly improve the initial commendable effort made by the authors. These problems in normative analysis, however, are not fatal to the manuscript in this reviewer's opinion. Ideally, a manuscript such as this further opens the minds of many privileged actors in the global health space to recognizing just how saturated through our work is with increasingly institutionalized inequity, and that radical reformation is called for if we want to avoid growing, legitimate indictments of moral hypocrisy.

Response: The reviewer is correct. As described in the Acknowledgements, although the authors sought review and input from a bioethicist throughout the development of this paper, the authors themselves do not claim to be experts in moral or political philosophy or ethics. Our goal was to highlight what we consider to be a serious concern, supported by empiric data by UTC offset on the global population, countries, human development index, and health. In other words, our intended audience is persons engaged in global health from a wide range of technical backgrounds, and our objective is to highlight an underappreciated problem in simple terms, accessible to a wide audience, so as to stimulate practical progress, or at least reflection on, global health teleconference scheduling. We of course hope that our work will foment theoretical progress by relevant experts with backgrounds in moral or political philosophy or ethics. That said, we very much agree that many considerations need to be brought to bear on our topic of focus besides merely 'equity of participation.' We have now edited the Discussion, Strengths and weaknesses of the study section to acknowledge this concern explicitly as 'We also acknowledge that factors beyond those contributing to equity of participation may be relevant in teleconference scheduling, but explicit consideration of these was beyond the scope of our analysis.'

3. Comment: Specific suggestions for revision: This reviewer found the abstract confusing to understand. The term "ethics of distribution" is vague and distinctly unhelpful as a simplified analytical construct to engage the reader; it needs elaboration and/or more precision in terms of what the authors' are trying to convey as a framework to guide their normative analysis. The abstract could simply state that the paper uses a specific, novel methodological strategy to map condensed time zones onto population health metrics to support a normative claim about how we might "most defensibly" schedule global health conferences where advancing equitable regard/treatment of would-be condition, relevant participants is the primary aim/goal. If the authors are interested in developing their normative analysis, and/or calling out a rich tradition of moral inconsistency in terms of the lofty moral rhetoric rarely matching the actual work within global health, that would also be most welcome.

Response: We agree that the Abstract could be clearer and we very much like the wording suggested by the reviewer. We have reworked the Abstract for clarity, including adopting the wording suggested. We do consider the term 'ethics of distribution' to be well founded in the ethics literature, but agree that it requires elaboration to be accessible to a broad audience. We have now elaborated on this term when first mentioned in the body of the manuscript, and at some points thereafter. We also agree that there is a rich tradition of moral rhetoric rarely matching practice in global health. While comprehensive treatment of this issue is beyond scope of our pragmatic analysis, we agree that it should be mentioned. We now address this issue in a new paragraph of the Discussion, under 'Implications for clinicians and policymakers.'

Reviewer: 2

Mr. Binyam Tariku, College of Health Science Dilla University

1. Comment: This paper describes approaches of scheduling teleconferences for multiple time zones, which is an important notion to address global health issue. I believe study findings will provides an

important stepping stone for alleviating the accessibility and equitability global telehealth conferences. However, it will be great if the authors includes implication and recommendation for future studies.

Response: The authors thank the reviewer for the positive feedback. We agree that recommendations for future work should be addressed. We have added the statement 'We suggest that practical tools be designed to display global population, countries, human development index, and health metrics by condition by UTC offset to inform global health teleconference scheduling' to the Conclusions, since the main issue remaining is application of our findings to teleconference scheduling decisions. Our group is already working on such software tools.

Reviewer: 3

Dr. Robert Lipton, Prevention Research Center

1. Comment: I really like this paper as it does something different in exploring disparities/inequities, globally. I particular like the inclusion of ethical considerations as well as some very interesting/novel graphics, particularly the globe! The DALY approach is also important.

Response: The authors are delighted that the reviewer appreciated the value of our work.

2. Comment: My sense, as well, is this is a bit of a trojan horse for larger, important disparities work and begs the question about focusing on specific cases, like comparing India to the UK or the US. And what do we do about Russia - Twelve times zones averaged into one!?!? I think this is a good first pass at dealing with such issues using this spatial approach but as it stands, It's a bit vulnerable to over averaging – ecological fallacy issues, to some extent.

Response: We agree that the worked examples might be perceived as a 'Trojan horse.' However, rather than being intentionally subversive, our purpose is to illustrate the problem using examples with which the authors are highly familiar, and that readers from a broad range of technical backgrounds could easily understand. We are open to removing the worked examples if they would be perceived to be an 'ambush' on particular groups or individuals. However, that was certainly not our intention and we do not interpret that the reviewer is necessarily requesting that they be removed.

We also agree that representing multiple time zone countries in a single time zone, or 'bunching,' is problematic and this concerned is expressed explicitly in the limitations section of the Discussion. However, we would highlight that we opted for the most parsimonious approach to this problem, that decisions were taken primarily for the purpose of illustration, and that our work could be improved by the somewhat daunting task of identifying sub-national population and health metrics by location and applying them to subnational UTC offsets. In reality, UTC offset is merely a surrogate for local apparent solar time. Local apparent solar time best captures whether the person is experiencing day or night, at what point they are in the diurnal cycle, and whether or not they are being asked to participate in a teleconference at an unfavorable point in their diurnal cycle. Thus, in the example provided for Russia, the most robust approach would be to consider the local apparent solar time at the location of each potential Russian participant and to incorporate this into decisions about teleconference scheduling. However, such an approach is highly computationally demanding with available tools, and so we proposed UTC offset as a pragmatic, albeit somewhat flawed, means of estimating 'inconvenience.' We expand the limitations section of the Discussion to describe UTC offset as a surrogate for local apparent solar time.

3. Comment: Also, in terms of the ethics discussion, it either needs more or less discussion, as does the colonized versus colonizer framework. Is there a hierarchy of ethical frameworks? Or do we really need to just default to egalitarian approaches, notwithstanding it may be "Insufficient to provide equity to disadvantaged groups" all things being equal. (btw, isn't this almost always going to be the case in this very imperfect world?). Also, what if the "center" (center versus periphery) contribution is absolutely essential to helping mitigate health disparities and thus should be accorded some importance (again, this is both in the realm of philosophical discussion and the reality of doing international health). Another issue, which is beyond the scope of this work has to do with the possibility that even with the temporal inequity, such teleconferences might be better than the alternatives. Some discussion of this may be appropriate. Also, clarification between conferences and meetings should be made and a brief discussion of costs and benefits.

Response: Our work on the topic of global health teleconference scheduling is transdisciplinary and as is typical of such work, it has been challenging to incorporate both a quantitative spatial exploration with an effort to support a normative ethics-based position on how to most defensibly schedule global health teleconferences with the primary goal of advancing equitable regard for participants by health condition. Reviewer 1 has raised a number of points in the domains of moral or political philosophy and ethics that have allowed us to expand the ethics discussion in the revised manuscript, so we have opted for 'more than less' in this regard to address the reviewers' points. Reviewers have also highlighted that there are obviously considerations for global health teleconference scheduling beyond the mere equity of participation that we now address explicitly in the revised manuscript. In addition, we have also expanded on the broader context of moral and ethical inconsistencies in global health in which the specific concerns highlighted in our analysis occur.

4. Comment: Another interesting issue is that the authors mention two decades of data collection which seems like something really important and demands a bit more explanation. We can imagine all kinds of really important information coming out of this project.

Response: Of the past two decades, the senior author has spent the first decade based in Tanzania and the second decade based in New Zealand. He is thus familiar with both the perspective of low- and middle-income country participation, and with participation very remote to zulu time in global health teleconferences. BMJ author requirements call for a 'Patient and public involvement' section in the Methods, but space is insufficient and journal convention does not provide for reporting of corresponding 'results' of what in our case is ultimately the qualitative experience of the authors. We have re-written the 'Patient and public involvement' section to be more specific about the nature of these formative experiences.

5. Comment: Small things: the shift worker – carcinogen connection seems a bit overstated for this work. Not really needed to make the larger argument.

Response: We agree that the carcinogen claim seems overstated and we have removed this from the narrative text. Instead we describe impacts on wellbeing and health more generally.

6. Comment: Small things: I do think the first figure with the timeline and world atlas needs to be higher resolution and/or larger, as it stands, it's a bit unreadable but very interesting.

Response: We have checked that the Figures meet and in fact exceed BMJ Open minimum dpi requirements and have reuploaded all image files. We will request that the Editor ensure that the main figure is published in 'full page' format to ensure optimal readability. We suspect that the problem is not with the resolution of the submitted image, but how it was served to reviewers.

VERSION 2 – REVIEW

REVIEWER	Lipton, Robert Prevention Research Center
REVIEW RETURNED	16-Mar-2022

GENERAL COMMENTS	This is a very unique paper and, notwithstanding some issues, after the revision, seems worthy of publication. I have issues with the outcome of interest, as it doesn't necessarily bear the weight of the mostly implicit claim that inequities in conference/meeting timing is related to inequities generally, which I think the authors really want to get at. There is really not research on the relationship between conferences/meetings and inequities, as far as I know. I do think this is a great methods paper, and I do like the inclusion of an ethics discussion, mostly as a start. I would, in the future, like to see this method applied to actual health outcome differences, using DALYS and the HDI, etc.
---